# Tunable Spacing Dual-Wavelength Q-Switched Fiber Laser Based on Tunable FBG Device

**Nurnazifah M. Radzi [1], Amirah A. Latif [1,\*], Mohammad F. Ismail [2], Josephine Y. C. Liew [1] , Noor A. Awang [3], Han K. Lee [4], Fauzan Ahmad [5], Siti F. Norizan [6] and Harith Ahmad [2]**

1   Department of Physics, Faculty of Science, Universiti Putra Malaysia, Serdang 43400, Malaysia; nazifahradzi95@gmail.com (N.M.R.); josephine@upm.edu.my (J.Y.C.L.)
2   Photonics Research Centre, University of Malaya, Kuala 50603, Malaysia; faizalis@um.edu.my (M.F.I.); harith@um.edu.my (H.A.)
3   Optical Fiber Laser Technology (OpFLAT) Focus Group, Department of Physics and Chemistry, Faculty of Applied Sciences and Technology, Universiti Tun Hussein Onn Malaysia, Pagoh 84600, Malaysia; norazura@uthm.edu.my
4   Department of Electronic Engineering, Faculty of Engineering and Green Technology, Universiti Tunku Abdul Rahman, Jalan Universiti Bandar Barat, Kampar 31900, Malaysia; hklee@utar.edu.my
5   Malaysia-Japan International Institute of Technology, Universiti Teknologi Malaysia, Jalan Sultan Yahya Petra, Kuala 54100, Malaysia; fauzan.kl@utm.my
6   Department of Physics, International Islamic University of Malaysia, Kuantan 25000, Malaysia; sitifn@iium.edu.my
\*   Correspondence: amirahlatif@upm.edu.my; Tel.: +603-9769-6985

**Abstract:** A tunable spacing dual-wavelength Q-switched fiber laser is experimentally demonstrated based on a fiber Bragg grating tunable device incorporated in an erbium-doped fiber laser (EDFL). The system utilizes two identical fiber Bragg gratings (FBGs) at 1547.1 nm origin to enable two laser lines operation. The wavelength separations between two laser lines are controlled by fixing one of the FBGs while applying mechanical stretch and compression to the other one, using a fiber Bragg grating tunable device. The seven steps of wavelength spacing could be tuned from 0.3344 to 0.0469 nm spacing. Pulse characteristics for both close and wide spacing of dual-wavelength Q-switched fiber laser are successfully being recorded. The findings demonstrate the latest idea of dual-wavelength fiber laser based on FBG tunable device, which offers a wide range of future applications.

**Keywords:** Q-switched fiber laser; erbium-doped fiber laser; dual-wavelength; tunable spacing; fiber Bragg gratings; carbon nanotubes

## 1. Introduction

Dual-wavelength fiber lasers, which are versatile light sources capable of providing more than single discrete laser emission lines, have sparked considerable interest in a wide range of applications, including remote sensing instruments [1,2] and fiber-based sensors [3–6], optical communication systems, microwave photonics [7–10], millimeter-wave [11] and terahertz waves [12,13], spectroscopy [14] and biomedical research [15]. Until now, many approaches for generating dual-wavelength lasing that implementing various techniques have been proposed, including utilization of comb filters with specialty fiber [16–18], phase modulator [19,20], nonlinear optical effects [21,22] and hybrid-gain schemes [23]. Having these techniques in a passively Q-switched fiber laser mechanism will generate dual laser lines with pulsed output. By using a tunable FBG device for Erbium-doped fiber (EDF) laser in this experiment, a new potential of dual-wavelength by passive optical devices can be introduced at the 1.55 μm telecommunication window.

Saturated absorbent (SA) is an essential component of passive pulse lasers. They are traditionally classified as artificial or natural SA. Natural SA has been in the spotlight for being able to beat the widely used semiconductor saturated absorber mirror (SESAM) prior

to the discovery of the material as SAs. Carbon nanotubes were among the first materials used as SA because they are cheap, easy to assemble and used, when compared to the conventional saturable absorber, SESAM. Furthermore, CNT is an ideal SA because of its pulse characteristics, such as low saturation power, fast optical response, high energy pulse yield, and having a good stability performance when incorporated in the fiber laser cavity [24]. Until now, CNT is yet the most encouraging and economical SA to be used to maintain a robust pulsed laser performance as well as being cheap and easy to fabricate compared to other materials. In addition, CNTs have been widely used in a variety of ways, ranging from the most basic, which involves inserting a thin film of CNT between two fibers [25], to the most advanced, which involves spraying a CNT solution or coating it onto D-shaped fibers [26] and micro tapered fibers [27]. In this project, we retain the sandwich technique of CNT thin films in our cavity design to maintain good stability performance from Q-switched output and focusing on the wavelength tunability of the dual-wavelength output which is still not much explored by other researchers.

There are now various approaches using specialty fibers as fiber filters to generate tunable spacing dual-wavelength lasing in their configuration, either with a presence of pulse or without pulse [28]. They obtained specific wavelength spacing between two controllable laser lines. In this study, we focus on prior work utilizing fiber Bragg grating as a wavelength filter, which may be classified into two types, one involving just one FBG and the other including two FBGs for dual-wavelength output design. Recently, Gao et al. proposed a tunable dual-wavelength fiber ring-cavity laser based on an FBG and DFB laser injection, a stable tunable dual-wavelength lasing with a wavelength spacing of 2.08 nm and a tuning range from 2.08 to 5.34 nm has been achieved [29]. Luo et al. and Feng et al. proposed dual-wavelength ring-cavity continuous-wave fiber lasers without pulse outputs, as compared to our proposed setup, where stable dual-wavelength laser outputs were achieved by changing the operating temperature of the FBGs [30] and by using a single FBG, exploiting its birefringence characteristic for the dual-wavelength output [31]. Ibarra et al. proposed a tunable dual-wavelength operation of an all-fiber thulium-doped fiber laser based on tunable fiber Bragg gratings, a stable dual-wavelength lasing with a wavelength spacing from 0.54 nm to 9 nm was achieved [32]. Another configuration of tunable dual-wavelength is demonstrated by Wang et al. [33], whereby a switchable and tunable wavelength-emitting status was achieved by a Bragg grating written in polarization-maintaining- fiber Bragg grating (PM-FBG) with a tuning range from 0.02 to 0.52 nm. On the other hand, Zalkepaly et al. also reported a dual-wavelength fiber laser, whereby the dual-wavelength output offered capable to be switched between two wavelengths only, which are at 1532 and 1533 nm, without a dual-wavelength tuning mechanism applied to the optical cavity [34]. We grasped the concept from previous works by designing our metal block model to manually control the strain given to the FBG. In this study, we achieved an ultranarrow pulse laser for dual-wavelength output utilizing the easiest approach to adjust the FBG at ambient temperature. We designed our metal block utilizing FBG and offer a simpler approach to manage the wavelength tunability by employing this configuration.

By using a specific method, these outputs need to be manipulated to meet the requirement for specific applications, especially for tunable dual-wavelength by controlling the wavelength separation between them. Two fiber Bragg gratings (FBG) at the same central wavelength are used to produce dual-wavelength in the cavity. The FBG acts as a wavelength selector due to its narrow band reflection. In this experiment, the wavelength for one of the FBG, which is denoted as FBG 1, is fixed, while another FBG, which is denoted as FBG 2, is tuned using superimposed FBGs by applying the FBG 2 with some mechanical stretch and compression. However, EDF is a homogeneous gain medium at room temperature and makes it difficult to generate more than one laser line due to strong mode competition happens in the cavity. To encounter this mode competition, cavity losses are created to allow more than one laser to be operated by adjusting different losses for two FBGs, which cover from 1547.12 to 1547.45 nm with seven steps of wavelength separations.

In this paper, we propose a new simple tunable dual-wavelength Q-switched fiber laser by using carbon nanotubes (CNT) as a saturable absorber (SA). The dual-wavelength tuning capability is successfully demonstrated with the presents of two FBGs incorporated together in the cavity configuration. The wavelength separation is controlled by using the loss control method for two FBGs with the help of a tunable FBG device. A tunable dual-wavelength fiber laser is successfully achieved with seven steps of wavelength separation with wavelength spacing from 0.0469 to 0.3344 nm is attained.

## 2. Carbon Nanotubes/Polyvinyl Alcohol Blends Preparation

Carbon nanotubes/polyvinyl alcohol (CNT/PVA) thin film was prepared via a simple polymer casting approach by referring to H. Ahmad et al. [35]. First, 1 g of PVA (Fluka, Mw = 61,000) was immersed into 25 mL of deionized water. The mixture was then heated to 60 °C under stirring conditions for an hour until the PVA was completely dissolved. Meanwhile, 5 mg of CNT (Nanostructured and Amorphous Materials Inc., Los Alamos, NM, USA, 95%) was dissolved into 20 mL of deionized water. The solution was sonicated for 30 min and then stirred for another 30 min. Then, the CNT solution was poured into the PVA solution and stirred for 10 min. A total of 1 mL of the mixture was then poured into a petri dish immediately and placed in a clean environment for 24 h. Finally, the homogenous CNT/PVA thin film was peeled off from the petri dish and was ready to be used as the saturable absorber.

## 3. Saturable Absorption of CNT Material

The twin detector method was used to obtain the results for material saturable absorption as in the previous work [36]. This method is a characterization technique that is used to obtain saturable absorption properties for all saturable absorber materials. In our work, CNT was the SA material utilized in the experiment. The experimental data of the material saturable absorption are shown in Figure 1, where the dotted mark represents the experimental value for the modulation depth of CNT thin film, while the linear line represents the fitting curve based on the characterization of the modulation depth for the twin detector method. The calculated values of the modulation depth ($\alpha_0$), the saturation intensity ($I_{sat}$) and the non-saturable absorption loss ($\alpha_{ns}$) can be obtained after fitting the formula in Equation (1) based on the modulation depth numerical study [37]:

$$\alpha(I) = \frac{\alpha_0}{1 + \frac{I}{I_{sat}}} + \alpha_{ns} \tag{1}$$

where $\alpha(I)$ is the intensity-dependent absorption coefficient obtained from calculated values based on the plotted graph. From the fitting, the modulation depth, saturation intensity, and non-saturable absorption loss of the device are estimated to be 16.4%, 0.025 MW/cm$^2$ and 83.6%, respectively. This indicates that CNT thin film has a robust saturable absorption property and is comparable to other materials [38–40].

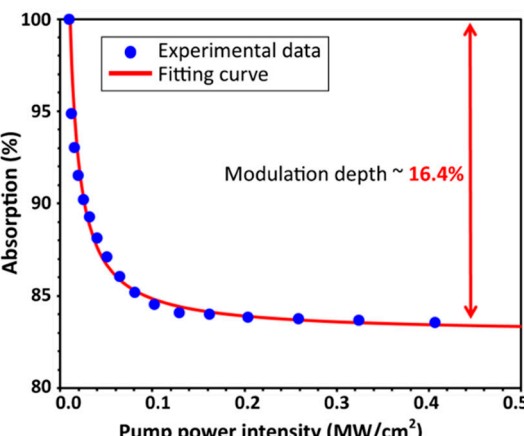

**Figure 1.** Measured saturable absorption data and its corresponding fitting curve for the CNT thin film as SA.

## 4. Experimental and Operation Principles

The experimental setup of tunable dual-wavelength Q-switched fiber laser by using CNT thin film is shown in Figure 2. The configuration setup had a laser diode (LD), a wavelength division multiplexer (WDM), a 3 m erbium-doped fiber (EDF) from Fibercore M-12 (with the absorption of about 18 dB/m), an isolator (ISO), a circulator, two optical couplers, and a CNT thin film as a saturable absorber (SA) working in the ring cavity. The EDF used in the experiment had a mode field diameter of 6.6 μm and a cut-off wavelength at 911 nm. The LD used was a Fiber Bragg Grating Stabilized LD, which acts as a 980 nm pump source with a maximum power of 400 mW. The LD pump source was injected into the laser cavity through a 980/1550 nm WDM through a forward pumping scheme. The injected light by the LD was then absorbed by a 3 m erbium-doped fiber which works as the gain medium of the cavity. The isolator ensures that the signal only propagates in one direction without reflecting back in the cavity. The SA consisted of two fiber ferrules sandwiched together with the CNT film attached between the two fiber ferrules. The circulator functioned to allow the signals to circulate and reflected back at port 2 where the two FBGs were located, as in Figure 2. The two FBGs were connected to a circulator function as a filter to the wavelength selection that was chosen to be emitted. The group velocity dispersion (GVD) of EDF in this cavity was 64 ps$^2$ km$^{-1}$, whereas the remainder of the SMF fiber (SMF-28) with a length of 7.01 m had an anomalous GVD of $-24$ ps$^2$ km$^{-1}$. The total net dispersion of this cavity was 0.024 ps$^2$. A 90:10 coupler was utilized to tap 10% of the lasing for further analysis, while the remaining 90% was connected back into the cavity. An OSA (Yokogawa AQ6370C Optical Spectrum Analyzer), an oscilloscope (RTM3002 Oscilloscope), an RFSA (FPC-1000 Radio-Frequency Spectrum Analyzer) and an optical power meter (OPM) were used to examine the 10% output. The oscilloscope was used to measure the laser pulses from the fiber laser and OSA as an analyzer to monitor the spectrum from the cavity.

Two FBGs were used in the experiment. Both of the FBGs were identical, reflecting at the central wavelength of 1547.12 nm in the ambient temperature (before applying any stretch and compressions). The wavelength reflectivity of the FBGs was 98%, with 0.32 nm of 3 dB reflection bandwidth. One of the FBGs, denoted as FBG 2, was connected to a tunable FBG device, as in Figure 3. The adjustable screws were turned to push the movable metal block to bend the flexible plate where the FBG was attached by using epoxy glue. At the same time, there some mechanical stretch and compressions were applied to the FBG. The FBG deformation lead to a displacement of the Bragg wavelength, which allowed for tuning of the wavelength of the generated laser line [41]. In this case, one of the FBGs was stationary, while the other one was being tuned by a tunable FBG device. The spacing wavelength was controlled by adjusting the tunable FBG device from wide to close spacing between two laser lines. The filtered dual-wavelength and that pulsed from

the cavity were generated simultaneously. The tunable dual-wavelength covered from 1547.1156 to 1547.4500 nm with seven steps of wavelength separations. However, EDF was a homogeneous gain medium at room temperature, thus made it difficult to generate more than one laser line due to strong mode competition occurring in the cavity [42]. To encounter this mode competition, cavity losses were created to allow more than one lasing emission to be operated by adjusting different losses for two FBGs [43,44]. The goal of producing losses is to minimize competition by adjusting intra-cavity losses. In our experiment, the intra-cavity losses were done by giving a sufficient bending loss at the fiber of the two FGBs to create the individual loss at the reflection of the FBGs. This technique reduced the dominant power that came from one of the dominant FBG's lasing, thus solving the mode competition issue.

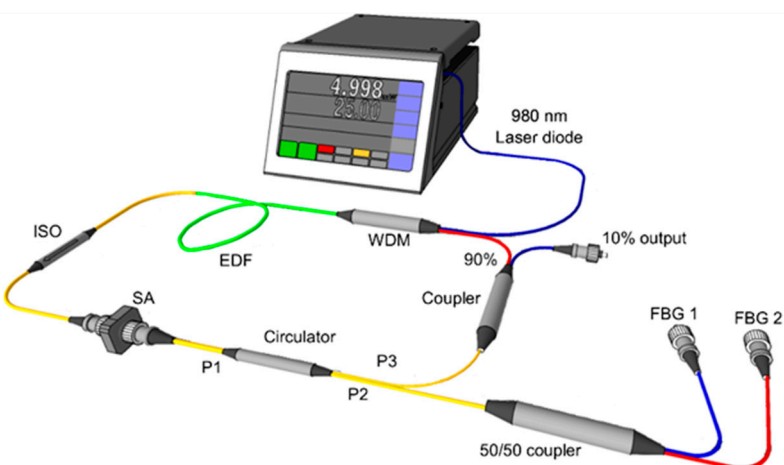

**Figure 2.** Tunable dual-wavelength of Q-switched fiber laser configuration.

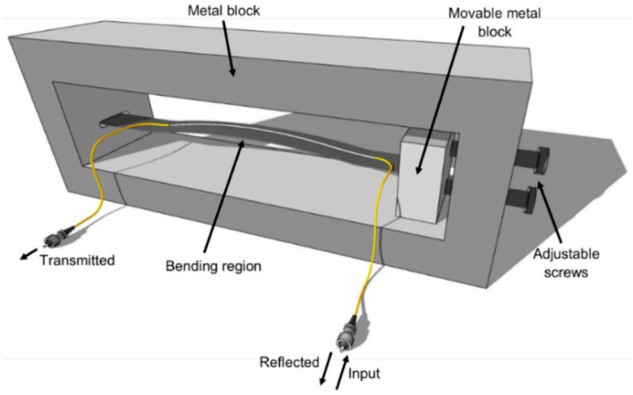

**Figure 3.** Tunable device fiber Bragg grating from FBG 2.

## 5. Results and Discussion

Figure 4 depicts the spectra of the ring cavity erbium-doped fiber laser when both FBGs are applied in the complete circuit. With the proposed configuration, a balance dual-wavelength laser output operation can be achieved effectively by adjusting the cavity losses to both FBGs used in the design.

The dual-wavelength Q-switched fiber laser output, as illustrated in this Figure 4, was initiated from the two FBGs that are being manipulated using the tunable FBG device. The red colour line at a shorter wavelength originated from the actual FBG's wavelength at 1547.12 nm (FBG 1). In contrast, the blue colour line at a longer wavelength is the wavelength that was tuned using the tunable FBG device at 1547.45 nm (FBG 2). The blue colour lasing represents the maximum wavelength tunability, since the stretch and compression of the FBG reached their limit. The dual-wavelength spectrum, represented by

the black line in the figure, appears from the combination of two laser lines, represented in the figure by the red and blue colour spectra. It was ensured that both FBGs were monitored at room temperature with no substantial temperature fluctuation. The dual-wavelength spectrum, indicated by the black line in the spectrum, results from the combining of two laser lines (red and blue). The distance between two lasing lines can be increased by regulating both FBGs with two metal blocks. The 3 dB line widths of the dual-wavelength output are 0.028 nm at 1547.12 nm (shorter wavelength) and 0.031 nm at 1547.45 nm (longer wavelength), coming from FBG 1 and FBG 2, respectively. The optical signal-to-noise ratio (OSNR) values of the dual-wavelength are 56.0 and 57.5 dB for the shorter wavelength and longer wavelength peaks, respectively.

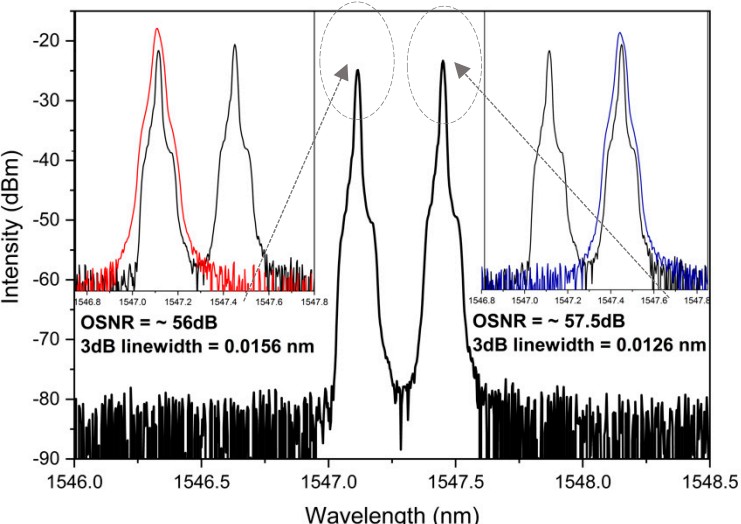

**Figure 4.** Dual−wavelength peak wavelength output from our proposed design, where the shorter wavelength comes from FBG 1, whilst the longer wavelength comes from FBG 2.

A stable spacing tunability of the dual-wavelength fiber laser is achieved by adjusting the tunable FBG device, as shown in Figure 5. The spacing wavelength is tuned continuously in seven steps increment from 0.0469 to 0.3344 nm at the maximum pump power of 90.88 mW. The maximum wavelength separation is limited by the damage of FBG under stretching conditions, but can be enhanced by improving the construction robustness of FBG, such as using good recoating material technology [45].

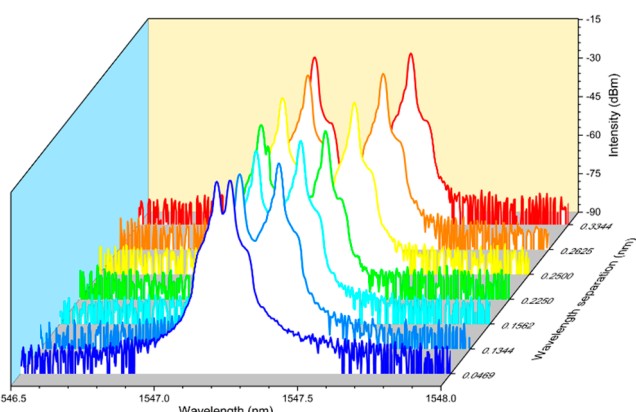

**Figure 5.** Output spectra for different wavelength separations of the dual-wavelength Q-switched fiber laser.

Figure 6a,b shows the output spectrum stability of the tunable dual-wavelength Q-switched fiber laser for a minimum spacing of 0.0469 nm and a maximum spacing of 0.3344 nm, taken at a fixed pump power of 90.88 mW. The stability measurement is taken

at room temperature for every 5 min in an hour of operating time. The system is capable of operating effectually, where it was observed that the dual-wavelength laser emission shows excellent output stability. Meanwhile, the characteristics of the pulses in Figure 6c,d show no difference, since both pulse trains operate at an equal pump power of 90.88 mW. These results proved that the behavior of the Q-switched fiber laser is obeyed since the frequency changes according to the increasing of the pump power [46–48].

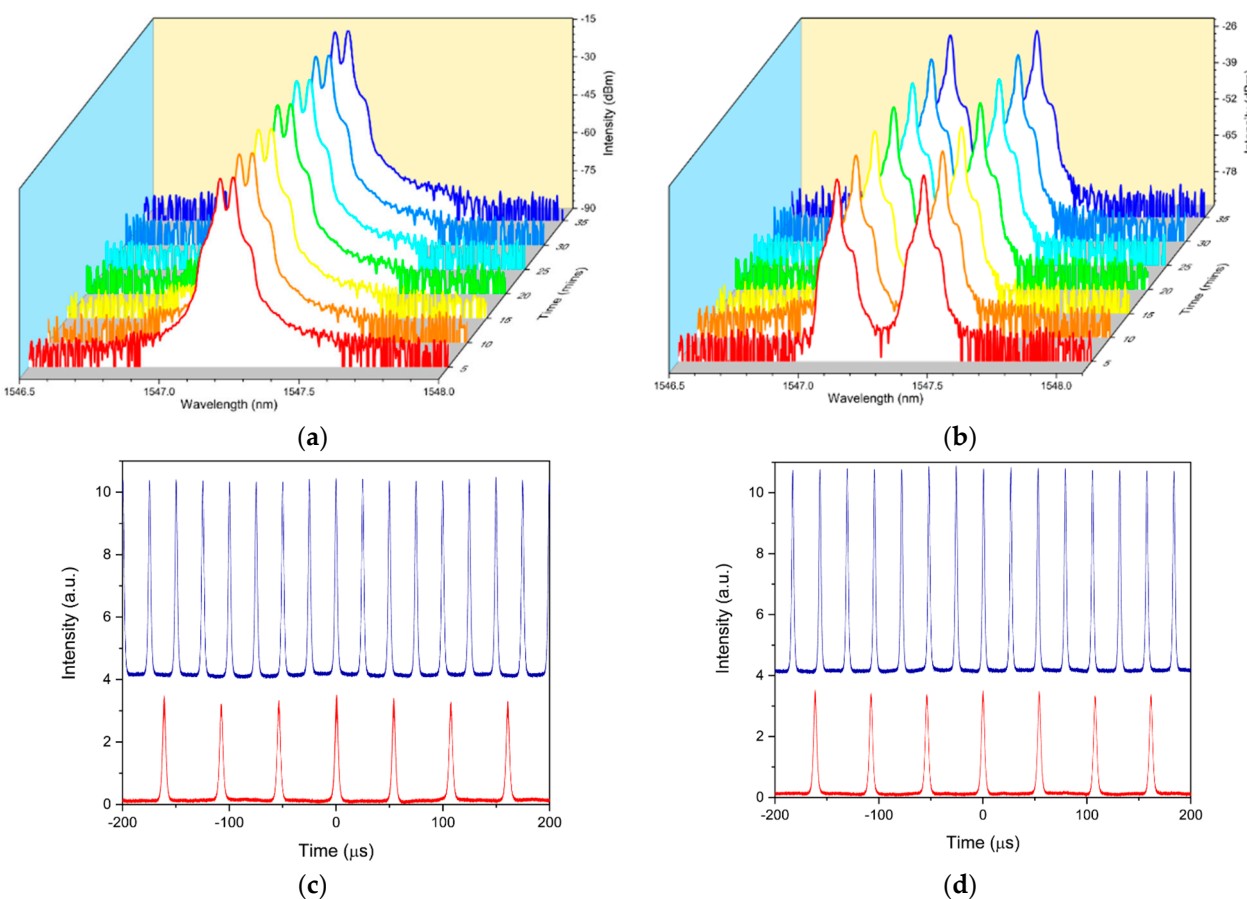

**Figure 6.** (**a**) Output spectra stability for close spacing; (**b**) output spectra stability for wide spacing; (**c**) pulse train for close spacing; (**d**) pulse train for wide spacing.

The pulse characteristics of the Q-switched fiber laser, such as pulse repetition rate and pulse width for both close and wide spacing, are plotted as in Figure 7. The increase of repetition rate from 18.6121 to 40.0664 kHz depends on the increasing pump power from 35.30 to 90.88 mW, with corresponding decreasing pulse width from 3.22 to 2.504 µs for close spacing dual-wavelength output. Whereas the increase of the repetition rate from 18.58 to 38.21 kHz depends on increasing the pump power from 35.30 to 90.88 mW, with corresponding decreasing pulse width from 3.29 to 2.31 µs for wide spacing dual-wavelength output. The nonlinear dynamics of the EDF and the SA are interconnected according to the dynamics of the energy provided by the pump power. As the pump power increases, more gain will result to saturate the SA. In addition, the threshold energy stored in the amplification medium (EDF) can also be reached more quickly to produce pulses. As a result, pulses are produced quicker. This, thus, brings about a decline in the pulse width, as the speed of the pulse creation becomes faster. On the other hand, increasing the pump power increases the repetition rate, as more longitudinal modes can pass the threshold of the Q-switching process and locked together, producing a higher repetition rate of the Q-switched output [49]. The pulse width obtained from the experiment could

be further narrowed by optimizing parameters such as cavity length, modulation depth or minimize the losses in the cavity [50].

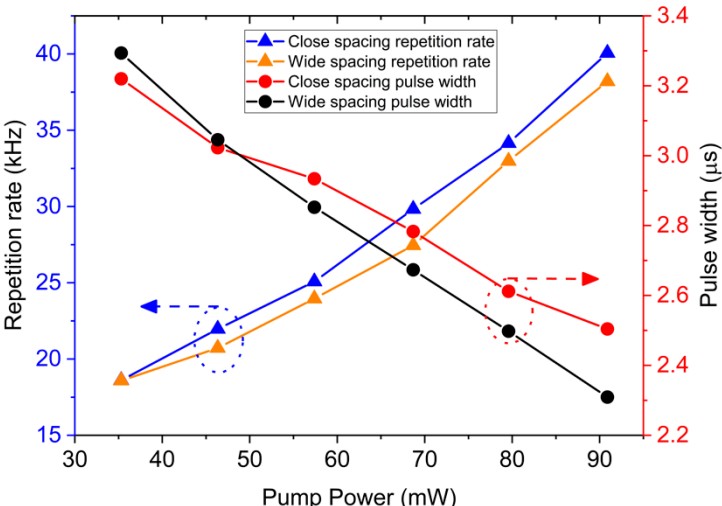

**Figure 7.** Pulse repetition rate and pulse width as a function of pump power.

Table 1 highlights the performance comparison among several tunable dual-wavelength fiber lasers based on FBG in this experiment. Several studies were being conducted on continuous-wave (CW) lasers rather than pulsed lasers, such as Q-switched and mode-locked fiber lasers. Tunable dual-wavelength pulse output designs mentioned in the table had more advantages as they provide higher lasing quality, which means they have a higher potential to be used in optical fiber sensors and wavelength converters in optical communication systems [51,52]. Based on previous work, the dual lasing element illustrated that tunable dual-wavelength may be created by changing the temperature to obtain tunable separation between two laser lines [53]. We suggested tunable spacing performance in room temperature based on strain and stretch application that may be used independently, since FBGs' sensitivity is higher towards strain rather than the temperature changes [54]. When assessing other research works previously [32,55], it could be seen that problems were encountered in terms of unstable dual-wavelength output from the FBGs. In this experiment, we successfully reported fine-tunable spacing by using the metal block for easy FBG tuning, with good stability performance, to enhance the tuning stability of the pulsed dual-wavelength output.

**Table 1.** Performance comparison of several tunable dual-wavelength fiber lasers based on FBG.

| Configuration | Laser Type | Dual Lasing Element | Technique | Tuning Range (nm) | Ref. |
|---|---|---|---|---|---|
| SOA | CW laser | $\lambda_1$ = FBG $\lambda_2$ = DFB laser | Temperature control | 2.08 to 5.34 nm | [29] |
| SOA | CW laser | $\lambda_1$ = cascaded FBG $\lambda_2$ = cascaded FBG | Temperature control | 0.18 to 0.6 nm | [30] |
| TDFL | CW laser | $\lambda_1$ = tunable FBG $\lambda_2$ = tunable FBG | Strain application | 1.7 to 3.7 nm | [32] |
| TDFL | CW laser | $\lambda_1$ = FBG $\lambda_2$ = translation FBG | Stretch application | 0 to 5.14 nm | [55] |
| YDF | Pulsed laser | PM-FBG | Adjusting polarization state | 0.02 to 0.52 nm | [33] |
| EDFL | Pulsed laser | $\lambda_1$ = FBG $\lambda_2$ = metal block FBG | Strain application | 0.0469 to 0.3344 nm | [This work] |

## 6. Conclusions

A simple, tunable dual-wavelength Q-switched fiber laser with CNT-SA was experimentally demonstrated. The tunable dual-wavelength output capability was tested by utilizing two mechanically stretched and compressed FBGs that serve as the tunable dual laser lines output. Dual-wavelength generation requires a fine adjustment of the cavity loss for both wavelengths. The seven steps of wavelength spacing could be tuned from 0.0469 to 0.3344 nm. The 3 dB linewidths of the dual-wavelength output were 0.028 nm at the shorter wavelength and 0.031 nm at the longer wavelength, from FBG 1 and FBG 2, respectively. The increase of repetition rate was in the range of 18.61 to 40.07 kHz for close-spacing dual-wavelength output and 18.58 to 38.21 kHz for the wide spacing dual-wavelength output, with a maximum pump power of 90.88 mW. Pulse characteristics for both close and wide spacing of the dual-wavelength Q-switched fiber laser were successfully conducted and presented in the proposed design.

**Author Contributions:** Conceptualization, A.A.L.; methodology, N.M.R. and F.A.; validation, H.A., F.A. and J.Y.C.L.; formal analysis, N.M.R., N.A.A. and S.F.N.; investigation, N.M.R. and M.F.I.; resources, N.M.R. and H.K.L.; data curation, N.M.R. and J.Y.C.L.; writing—original draft preparation, N.M.R.; writing—review and editing, A.A.L.; supervision, A.A.L., N.A.A. and H.A. All authors have read and agreed to the published version of the manuscript.

**Funding:** This research work was funded by the Fundamental Research Grant Scheme (FRGS) under grant number FRGS/1/2018/STG02/UPM/02/1/5540123 from the Ministry of Higher Education Malaysia.

**Institutional Review Board Statement:** Not applicable.

**Data Availability Statement:** Not applicable.

**Conflicts of Interest:** The authors declare no conflict of interest.

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
