# Peer review of "Tunable Spacing Dual-Wavelength Q-Switched Fiber Laser Based on Tunable FBG Device"

_photonics, doi:10.3390/photonics8120524_

Round 1

Reviewer 1 Report

Dear Authors, This is a quite interesting experimental paper. Photos are useful for the readers and they are quite explicative

The topic of the paper is suitable for Photonics
I would suggest to perform a revision and an editing by an english native speaker. The paper is sometimes difficult to follow   In the results section (page 6) I would increase data regarding operating system and device power stability    In section 6 I would soften the conclusion, changing as follows "The tunable dual wavelength output capability has been tested by utilizing two mechanically stretched and compressed FBGs that serve as the tunable dual laser lines output"

Author Response

REVIEWER 1

The topic of the paper is suitable for Photonics.

(1) I would suggest to perform a revision and an editing by an english native speaker.

ANSWER:

Revision and editing by an English native speaker have been carried out for the revision of this manuscript.

_________________________________________________________________________________

(2) The paper is sometimes difficult to follow   In the results section (page 6) I would increase data regarding operating system and device power stability.

ANSWER:

The section has been made considering how the setup operates by describing the FBG tunable device to change the output wavelength, and subsequently stability when the dual wavelength output is changed at close spacing and wide spacing of the dual wavelength output. Then this section also explains about pulse train, repetition rate and pulse width. This arrangement is appropriate for this manuscript because this method of presentation is also used by other articles. however, we are really grateful for the opinion. The reviewer's opinion has been considered by changing the way of explanation in this writing to provide a better understanding to the readers.

_________________________________________________________________________________

(3)  In section 6 I would soften the conclusion, changing as follows "The tunable dual wavelength output capability has been tested by utilizing two mechanically stretched and compressed FBGs that serve as the tunable dual laser lines output"

ANSWER:

Thank you for the reviewer's comment. The correction has been made to this manuscript for a clearer view to the readers.

_________________________________________________________________________________

Reviewer 2 Report

This paper presents a tunable dual-wavelength Q-switched fiber laser using two FBGs, and with carbon nanotubes as a saturable absorber. The tunable fiber ring laser with dual wavelengths using FBGs are well investigated in the literature. Overall, I don’t feel this manuscript has any novelty compared to the literature demonstrations. Even though, there is a lack of information on operating principles and laser wavelength/power stability over time. I did not find this analysis to make any originality, and not suitable to publish in Photonics.

1. N.U.H.H. Zalkepali, N.A. Awang, A.A. Latif, Z. Zakaria, Y.R. Yuzaile, N.N.H.E.N. Mahmud, “Switchable dual-wavelength Q-switched fiber laser based on sputtered indium tin oxide as saturable absorber”, Results in Physics, vol. 17, p. 103187, 2020.

2. X.Feng, L. Sun, L. Xiong, Y. Liu, S. Yuan, G. Kai, X. Dong, “Switchable and tunable dual-wavelength erbium-doped fiber laser based on one fiber Bragg grating”, Optical Fiber Technology, vol. 10, pp. 275-282, 2004.

Author Response

REVIEWER 2

(1) This paper presents a tunable dual-wavelength Q-switched fiber laser using two FBGs, and with carbon nanotubes as a saturable absorber. The tunable fiber ring laser with dual wavelengths using FBGs are well investigated in the literature. Overall, I don’t feel this manuscript has any novelty compared to the literature demonstrations.

  1. N.U.H.H. Zalkepali, N.A. Awang, A.A. Latif, Z. Zakaria, Y.R. Yuzaile, N.N.H.E.N. Mahmud, “Switchable dual-wavelength Q-switched fiber laser based on sputtered indium tin oxide as saturable absorber”, Results in Physics, vol. 17, p. 103187, 2020.
  2. X.Feng, L. Sun, L. Xiong, Y. Liu, S. Yuan, G. Kai, X. Dong, “Switchable and tunable dual-wavelength erbium-doped fiber laser based on one fiber Bragg grating”, Optical Fiber Technology, vol. 10, pp. 275-282, 2004.

ANSWER:

Thanks for the reviewer's comments. However, we sincerely hope that this article can be accepted for publication as there is a novelty in this manuscript. This is because we propose a new simple tunable dual-wavelength Q-switched fiber laser by using CNT as a saturable absorber. The tunability is being achieved by using the FBG tunable device incorporated together in the ring cavity. A few articles reporting tunability of dual wavelength Q-switched output previously are not using this FBG tunable device as what we are proposing. As the article showed by the reviewer, the first article, in this reference,

 N.U.H.H. Zalkepali, N.A. Awang, A.A. Latif, Z. Zakaria, Y.R. Yuzaile, N.N.H.E.N. Mahmud, “Switchable dual-wavelength Q-switched fiber laser based on sputtered indium tin oxide as saturable absorber”, Results in Physics, vol. 17, p. 103187, 2020.

In this article, it is shown that work done by Zalkepali et al. is a switchable dual-wavelength Q switched fiber laser. The configuration is different from the one we are proposing. In addition, the switching capability is limited since the tuning of the Fiber Bragg Grating (FBG) was not being done to have a close and wide spacing of the tunable dual-wavelength output. They are only using the switching mechanism to make the laser operates either from 1532 nm or 1533 nm wavelength output. Another difference is that we use CNT as our saturable absorber compared to the one that they used i.e. indium tin oxide.

Another  previous article pointed out by the reviewer is as below,

X.Feng, L. Sun, L. Xiong, Y. Liu, S. Yuan, G. Kai, X. Dong, “Switchable and tunable dual-wavelength erbium-doped fiber laser based on one fiber Bragg grating”, Optical Fiber Technology, vol. 10, pp. 275-282, 2004.

In this article, it is shown that the work done by Feng et al. is switchable and tunable dual-wavelength fiber laser output by using fiber Bragg grating. However, this article is not presenting the Q-switched dual-wavelength output or at least pulsed dual wavelength output. They were only reported on the continuous-wave dual wavelength output. This characteristic made a huge difference in the article compared to the one we are proposing. This is due to the pulsed output obtained from the Q-switched dual-wavelength fiber laser giving extra properties that can be extracted from the time domain measurement, especially for optical fiber sensors and Light detection and ranging (LiDAR) applications.

_________________________________________________________________________________

 (2)Even though, there is a lack of information on operating principles and laser wavelength/power stability over time. I did not find this analysis to make any originality, and not suitable to publish in Photonics.

ANSWER:

Thank you for the reviewer's comment. The correction has been made to this manuscript for a clearer view to the readers. The reasoning to strengthen the novelty of our proposed article has been put forward in this article and we sincerely hope that this article can be accepted for publication in the Photonics Journal. Thank you.

A few added statements are as follows:

“Saturated absorbent (SA) is an essential component of passive pulse lasers. They are traditionally classified as artificial or natural SA. Natural SA has been in the spotlight for being able to beat the widely used semiconductor saturated absorber mirror (SESAM) prior to the discovery of the material as SAs. Carbon nanotubes were among the first materials used as SA because they are cheap, easy to assemble and used, when compared to the conventional saturable absorber, SESAM. Furthermore, CNT is an ideal SA because of its pulse characteristics such as low saturation power, fast optical response, high energy pulse yield, and having a good stability performance when incorporated in the fiber laser cavity [24]. Until now, CNT is yet the most encouraging and economical SA to be used to maintain a robust pulsed laser performance as well as being cheap and easy to fabricate compared to other materials. In addition, CNTs have been widely used in a variety of ways, ranging from the most basic, which involves inserting a thin film of CNT between two fibers [25], to the most advanced, which involves spraying a CNT solution or coating it onto D-shaped fibers [26] and micro tapered fibers [27]. In this project, we retain the sandwich technique of CNT thin films in our cavity design to maintain good stability performance from Q-switched output and focusing on the wavelength tunability of the dual-wavelength output which is still not much explored by other researchers.”

And

“We suggested tunable spacing performance in room temperature based on strain and stretch application that may be used independently since the FBGs’ sensitivity is higher towards strain rather than the temperature changes [52]. Based on other research works previously [31] [53], they encounter problems with unstable dual-wavelength output comes from the FBGs. In this experiment, we successfully reported fine-tunable spacing by using the metal block for easy FBG tuning with good stability performance to enhance the tuning stability of the pulsed dual-wavelength output. ”

And also,

They successfully reported their work with tunable Q-switched pulses. We grasped the concept from previous works by designing our metal block model to manually control the strain given to the FBG. In this study, we achieved an ultranarrow pulse laser for du-al-wavelength output utilizing the easiest approach to adjust the FBG at ambient temper-ature. We designed our metal block utilising FBG and offer a simpler approach to manage the wavelength tunability by employing this configuration. “

________________________________________________________________________________

Reviewer 3 Report

The paper entitled as "Tunable spacing dual wavelength Q-switched fiber laser based on tunable FBG device", by N. M. Radzi et al. presents a simple tunable dual-wavelength Q-switched fiber laser by using carbon nanotubes as a saturable absorber. The topic is fair, the experimental results are in good agreement with the theoretical ones, and the results show a sufficient novelty, and therefore, I suggest to accept the paper pending the revision:

  1. All abbreviations should be defined, for example in line 77 (PVA: polyvinyl alcohol)
  2. A brief comparison for the results is recommended (comparison between the present results and previous works such as [28]).
  3. The Introduction is well written and for fiber-based sensor (line 32) the recent works are suggested: 10.1016/j.sbsr.2021.100401; 10.1007/s11082-021-02969-x.
  4. It is mentioned that 10% output is needed to analyze the data through an optical spectrum analyzer, an oscilloscope, a radio-frequency spectrum analyzer, and optical power meter. This value (10%) could have been selected less. Please explain the selecting this value.
  5. There are some typo/grammatical error and the paper needs to be edited.
  6. Line 22 should be read as "tuned from 0.0469 nm to 0.3344 nm spacing. Pulse ....."

Author Response

REVIEWER 3

The paper entitled as "Tunable spacing dual wavelength Q-switched fiber laser based on tunable FBG device", by N. M. Radzi et al. presents a simple tunable dual-wavelength Q-switched fiber laser by using carbon nanotubes as a saturable absorber. The topic is fair, the experimental results are in good agreement with the theoretical ones, and the results show a sufficient novelty, and therefore, I suggest to accept the paper pending the revision:

  1. All abbreviations should be defined, for example in line 77 (PVA: polyvinyl alcohol)

ANSWER:

Thank you for the reviewer's comment. The correction has been made to this manuscript by adding definition to a few terms such as polyvinyl alcohol as PVA and polarization-maintaining- fiber Bragg grating as PM-FBG   for a clearer view to the readers.

_________________________________________________________________________________

  1. A brief comparison for the results is recommended (comparison between the present results and previous works such as [28]).

            ANSWER:

Thanks for the reviewer's comment. For reference in [28], the twin detector method is a characterization technique that is always being used to obtain saturable absorption properties for all saturable absorber materials. In our work, CNT is the SA material utilized in the experiment. A comparison between the technique we implemented and the technique on reference [28] was not made because both of the techniques are just the same. This reference is included in the manuscript to provide further information to the readers. However, amendments have been made to this manuscript to provide a clearer understanding to the readers. The additional statements are added to the manuscript as per the description underlined below:

“The twin detector method is used to obtain the results for material saturable absorption as in the previous work [28]. This method is a characterization technique that is used to obtain saturable absorption properties for all saturable absorber materials. In our work, CNT is the SA material utilized in the experiment.

________________________________________________________________________________

  1. The Introduction is well written and for fiber-based sensor (line 32) the recent works are suggested: 10.1016/j.sbsr.2021.100401; 10.1007/s11082-021-02969-x.

ANSWER:

Thank you for the reviewer's comment. Both of the references from: 10.1016/j.sbsr.2021.100401; 10.1007/s11082-021-02969-x.which belong to reference ; Sardar, M.; Faisal, M.; Ahmed, K. Simple hollow Core photonic crystal Fiber for monitoring carbon dioxide gas with very high accuracy. Sens. Bio-Sens. Res. 2021, 31, 100401.” And “Arman, H.;Olyaee, S. Realization of low confinement loss acetylene gas sensor by using hollow-core photonic bandgap fiber. Opt Quantum Electron. 2021,53, 1–13.” have been added to the reference lists as in ref. [3] and [4], respectively.

_________________________________________________________________________________

  1. It is mentioned that 10% output is needed to analyze the data through an optical spectrum analyzer, an oscilloscope, a radio-frequency spectrum analyzer, and optical power meter. This value (10%) could have been selected less. Please explain the selecting this value.

ANSWER:

Thank you for the reviewer's comment. We do agree with the reviewer that an output less than 10% could lead to the cavity optimization, in terms of more feedback energy given to the cavity. We chose this 10 % output portion due to the limited availability of different output portion of couplers in our laboratory. However, we do believe that the output would not affect significantly as far as our concern based on our previous experience. The output given by 10% portion of the coupler also gives an appropriate amount of power to be analysed by the measuring instrument used in this experiment.

________________________________________________________________________________

  1. There are some typo/grammatical error and the paper needs to be edited.

ANSWER:

Thank you for the reviewer's comment. The correction has been made to this manuscript for a clearer view to the readers.

_________________________________________________________________________________

  1. Line 22 should be read as "tuned from 0.0469 nm to 0.3344 nm spacing. Pulse ....."

ANSWER:

Thank you for the reviewer's concern. The correction has been made to this manuscript as below,

The seven steps of wavelength spacing could be tuned from 0.0469 nm to 0.3344 nm”, which has been underlined in the corrected version of the proposed manuscript.

_________________________________________________________________________________

Round 2

Reviewer 2 Report

The authors addressed all comments well, and now the manuscript has been improved with clear motivation, operating principles, and contrast from the previous works. 

  1. Show the SNR, and 3-dB linewidth values of both spectrums in Figure. 4.
  2. The Conclusion section needs to be expanded by added explicit information. 
  3.  I still suggest adding few sentences, what are the proposed configuration different from these references. 
  • N.U.H.H. Zalkepali, N.A. Awang, A.A. Latif, Z. Zakaria, Y.R. Yuzaile, N.N.H.E.N. Mahmud, “Switchable dual-wavelength Q-switched fiber laser based on sputtered indium tin oxide as saturable absorber”, Results in Physics, vol. 17, p. 103187, 2020.
  •  X.Feng, L. Sun, L. Xiong, Y. Liu, S. Yuan, G. Kai, X. Dong, “Switchable and tunable dual-wavelength erbium-doped fiber laser based on one fiber Bragg grating”, Optical Fiber Technology, vol. 10, pp. 275-282, 2004.
  •  

Author Response

We are really thankful for the comments from the reviewer. This constructive review is greatly appreciated as it has made this manuscript of better quality and better for all to read. Corrections and answers based on the questions given by the reviewer are as follows:

The authors addressed all comments well, and now the manuscript has been improved with clear motivation, operating principles, and contrast from the previous works. 

  1. Show the SNR, and 3-dB linewidth values of both spectrums in Figure. 4.

ANSWER:

Thank you for the reviewer's comment. The corrections have been made to this manuscript by adding the 3 dB linewidth and OSNR values of the dual-wavelength pulsed output in as illustrated in Figure 4.

Figure 4. Dual-wavelength peak wavelength output from our proposed design, where the shorter wavelength comes from FBG 1, whilst the longer wavelength comes from FBG 2.

With a few statements have been added to the manuscript as below:

“The distance between two lasing lines can be increased by regulating both FBG with two metal blocks. The 3dB linewidths of the dual-wavelength output are 0.028 nm at 1547.12 nm (shorter wavelength) and 0.031 nm at 1547.45 nm (longer wavelength) comes from FBG 1 and FBG 2, respectively. The optical signal to noise ratio (OSNR) values of the dual-wavelength are 56.0 dB and 57.5 dB, for the shorter wavelength and longer wavelength, respectively.”

  1. The Conclusion section needs to be expanded by added explicit information. 

ANSWER:

Thank you for the reviewer's comment. The corrections have been made to this manuscript by mentioning the two reported articles. A few statements have been added to the manuscript as below in the conclusion section as below:

The seven steps of wavelength spacing could be tuned from 0.0469 nm to 0.3344 nm. The 3 dB linewidths of the dual-wavelength output are 0.028 nm at the shorter wavelength and 0.031 nm at the longer wavelength, comes from FBG 1 and FBG 2, respectively. The in-crease of repetition rate is in the range of 18.61 kHz to 40.07 kHz for close-spacing du-al-wavelength output and 18.58 kHz to 38.21 kHz for the wide spacing dual-wavelength output, with the maximum pump power given of 90.88 mW. Pulse characteristics for both close and wide spacing of dual-wavelength Q-switched fiber laser are successfully being conducted and presented in the proposed design.”

  1.  I still suggest adding few sentences, what are the proposed configuration different from these references. 
  • N.U.H.H. Zalkepali, N.A. Awang, A.A. Latif, Z. Zakaria, Y.R. Yuzaile, N.N.H.E.N. Mahmud, “Switchable dual-wavelength Q-switched fiber laser based on sputtered indium tin oxide as saturable absorber”, Results in Physics, vol. 17, p. 103187, 2020.
  •  X.Feng, L. Sun, L. Xiong, Y. Liu, S. Yuan, G. Kai, X. Dong, “Switchable and tunable dual-wavelength erbium-doped fiber laser based on one fiber Bragg grating”, Optical Fiber Technology, vol. 10, pp. 275-282, 2004.

ANSWER:

Thank you for the reviewer's comment. The corrections have been made to this manuscript by mentioning the two reported articles. A few statements have been added to the manuscript as below:

 “Luo et al. and Feng et al. proposed dual-wavelength ring-cavity continuous-wave fiber lasers without pulse outputs, as compared to our proposed setup, where stable dual-wavelength laser outputs were achieved by changing the operating temperature of the FBGs [30] and by using a single FBG, exploiting its birefringence characteristic for the dual-wavelength output [31].”

And also

“On the other hand, Zalkepaly et al. also reported a dual-wavelength fiber laser, whereby the dual-wavelength output offered capable to be switched between two wavelengths only, which are at 1532 nm and 1533 nm, without a dual-wavelength tuning mechanism applied to the optical cavity [34].”

Both of the two references also have been added to the reference lists;

[31]  Feng, X.; Sun, L.; Xiong, L.; Liu, Y.; Yuan, S.; Kai, G.; Dong, X. Switchable and tunable dual-wavelength erbium-doped fiber laser based on one fiber Bragg grating. Optical Fiber Technology 2004, 10, 275-282.

[34]  Zalkepali, N.U.H.H.; Awang, N.A.; Latif, A.A.; Zakaria, Z.; Yuzaile, Y.R.; Mahmud, N.N.H.E.N. Switchable dual-wavelength Q-switched fiber laser based on sputtered indium tin oxide as saturable absorber. Results in Physics 2020, 17, 103187.

__________________________             ***THE  END***_________________________________
